# Role of Glucosinolates in the Nutraceutical Potential of Selected Cultivars of *Brassica rapa*

**DOI:** 10.3390/foods10112720

**Published:** 2021-11-07

**Authors:** Tania Merinas-Amo, María-Dolores Lozano-Baena, Sara Obregón-Cano, Ángeles Alonso-Moraga, Antonio de Haro-Bailón

**Affiliations:** 1Department of Genetics, Gregor Mendel Building, Faculty of Science, Campus Rabanales, University of Córdoba, 14014 Córdoba, Spain; b72lobam@uco.es (M.-D.L.-B.); ge1almoa@uco.es (Á.A.-M.); 2Department of Plant Breeding, Institute of Sustainable Agriculture, CSIC, Avd. Menéndez Pidal s/n, 14004 Córdoba, Spain; saraobregon@ias.csic.es (S.O.-C.); adeharobailon@ias.csic.es (A.d.H.-B.)

**Keywords:** turnip greens, gluconapin, progoitrin, health, safety, toxicity, chemoprevention

## Abstract

*Brassica rapa* L. subsp. *rapa* (turnip greens), a traditionally consumed vegetable, is well-known due to its high content of glucosinolates, which are secondary metabolites with a positive biological activity for human health. Our hypothesis has been based on the relation between *B. rapa* glucosinolate content and its healthy properties, and our aim is to establish guidelines for safe *B. rapa* vegetable consumption. Three *B. rapa* cultivars (143N5, 143N7 and 163N7) have been characterized by HPLC analysis of purified extracts from leaf samples in order to determine their glucosinolate content and to relate this content to beneficial effects on DNA protection, lifespan extension and chemoprevention. In order to ascertain the heath properties in vitro and in vivo, toxicity activities were assayed in the *Drosophila melanogaster* and leukaemia cell models; genomic safety was also assessed in both models using genotoxicity, fragmentation and comet assay. The *Drosophila* model has also been used to study the antioxidative activity and the longevity induction. Our results showed a relationship between *B. rapa* glucosinolate content and its safety and benefices in its consumption. Gluconapin, the main *B. rapa* glucosinolate, was directly related with these wholesome effects. The relevant conclusion in the present research is focused on *B. rapa* cultivar 163N7 due to its high gluconapin content and low progoitrin content, which exert anti-cancer and DNA protection properties and could be recommended as being safe and healthy for human consumption.

## 1. Introduction

Food, health and the environmental factors are intimately linked, it being necessary to establish a balance between the elements of this so called “health trilemma” for a healthier life on a more sustainable planet. The consumption of plant-based foods with nutraceutical properties is one of the crucial factors contributing to well-being, and to the promotion of health, to prevent various pathologies such as diabetes, cancer and cardiovascular and neurodegenerative diseases [1,2].

Several epidemiological and pharmacological studies have shown that the consumption of a diet rich in cruciferous vegetables (*Brassicaceae* family) may have an important role in protection from many chronic illnesses, including: cardiovascular disease, diabetes (II), dementia, age-related macular degeneration, immune dysfunction, obesity and some cancers [3,4,5].

Plants belonging to different species of *Brassica* are consumed all over the world, with the main consumers being found in China, India, Russia, Korea and Japan [6]. The importance of these vegetables comes from their high content of antioxidant components (vitamin C and phenolic compounds) and other specific health promoting compounds of the *Brassicaceae* family (GLS) [7]. In particular, protection against degenerative diseases from GLS and their enzymatic degradation products has been studied for decades and has been extensively reviewed [8,9,10,11,12,13].

*Brassica rapa* L. grows naturally from the western Mediterranean region to Central Asia. A wide range of its morphotypes has been described, including oilseed, leafy vegetables, root vegetables and fodder crops [14]. Its wide availability made it, several millennia ago, probably the first domesticated *Brassica* to be a multipurpose crop, and it has been widely used by all the civilizations evolving in that extensive region. It was cited in Sanskrit literature under the name of Siddharta [15].

Glucosinolates (GLS) are a large group of sulphur-containing compounds and are the principal secondary metabolites of the *Brassicaceae* family.

Naturally occurring GLS are (Z)-N-hydroximinosulfate esters, possessing a sulfur-linked β-D-glucopyranose moiety and amino acid-derived side chain. The structure of the side chain (R) is highly variable and may possess aliphatic, aromatic or indolyl groups. Intact GLS are chemically stable and relatively biologically inactive. Following disruption of the plant cell walls and organelles that contain them, the GLS are released. Enzymatic hydrolysis of the thioglucosidic bond by myrosinase (a β-thioglucosidase that co-occurs in plants with GLS) leads to different breakdown products with highly diverse biological activities [16]. Some GLS have been seen to have healthy activities, while others are harmful because they are potentially goitrogenic [17]. Their chemopreventive activity has been related to their metabolic breakdown products, mainly isothiocyanates (ITC). GLS/ITC intake has a health protective effect against different diseases or inflammatory processes [18]. Nevertheless, the GLS content is variable depending on many genetic and environmental factors (growing conditions, storage conditions and culinary treatments), making it difficult to determine its healthy effects in humans [19,20,21,22].

In northwest Spain and Portugal, there is a long tradition of cultivating *B. rapa* subsp. *rapa* to obtain turnips, turnip greens and turnip tops (called “grelos”), that are part of many dishes in the traditional cuisine of this area (for example “Lacón con grelos” or “Portuguese green soup”) [19].

In the Mediterranean diet, which is based on a high consumption of fruits and vegetables, these crops and products ought to occupy a prominent place, thus expanding their usual consumption area. In recent years, a breeding programme aiming to select *B. rapa* cultivars capable of growing in southern Spain, and producing turnip greens and turnip tops owing to improved agronomic and nutritional characteristics, has been undertaken. In the evaluation and selection process, the production capacity of turnip greens and turnip tops as well as the glucosinolate content of the harvested products were considered as the quality criteria of the final product. The most promising cultivars for breeding objectives would be those with the highest content of glucosinolates, which is related to beneficial effects [19,23]. In line with these criteria, three cultivars of *Brassica rapa* L. subsp. *rapa* were selected according to their good agronomic characteristics and their variability in glucosinolate content under Mediterranean climate growing conditions.

The objectives of this work were: (1) to assign to these selected cultivars some biological insight related to genomic safety/protection, life extension and anti-tumoural activities assayed in human and animal models, and (2) to relate the benefits of the consumption of these cultivars to their glucosinolate composition.

## 2. Materials and Methods

### 2.1. Plant Materials and Chemicals

Three *Brassica rapa* L. *subsp. rapa* cultivars were selected for this work: *B. rapa* 143N5, *B. rapa* 143N7 and *B. rapa* 163N7. The genetic origin of these cultivars is the *Brassica rapa* kept at the *Brassica* Germplasm Bank at the Biological Mission of Galicia (Pontevedra, Spain). From this germplasm collection, cultivars *B. rapa* 143N5, *B. rapa* 143N7 and *B. rapa* 163N7 were obtained at the Institute for Sustainable Agriculture (IAS, Córdoba, Spain) after several cycles of genetic selection for agronomic performance and variability in glucosinolate composition under Mediterranean climate conditions.

Plants were grown on an experimental farm at the IAS (N 37°8′, W 4°8′), where the climate is typically Mediterranean, with an average annual rainfall of 650 mm. The leaves of six plants per cultivar were harvested when they reached the optimal consumption stage. The five upper leaves per plant were sampled (tender leaves traditionally used for human consumption), weighed, frozen (24 h at −80 °C) and lyophilized with a freeze-drier Telstar model Cryodos-50 (Telstar, Terrasa, Spain). After lyophilisation, dry material was ground in a Janke and Kunkel Model A10 mill (IKA-Labortechnik, Staufen, Germany). The powder was then mixed and kept in air- and light-free conditions at low temperature (10 °C) until use.

Gluconapin (C_11_H_18_NO_9_S_2_) was purchased from PhytoPlan^®^ Heidelberg, Germany, 3417.9) and myrosinase (EC 3.2.1.147) from Sigma-Aldrich (St. Louis, MO, USA, T4528).

### 2.2. Extraction and Determination of Glucosinolates

The GSL levels of *B. rapa* leaf samples were determined by High Performance Liquid Chromatography (HPLC). Extraction and desulphation of glucosinolates from the freeze-dried samples was performed according to the method developed by Font et al. [24]. HPLC columns, solvents and gradient were fixed according to the ISO protocol (ISO 9167-1, 1992). An amount of 100 mg dry weight of lyophilized sample was precisely weighed, and a two-step glucosinolate extraction was carried out in a water bath at 75 °C to inactivate myrosinase. In the first step, the sample was heated for 15 min in 2.5 mL 70% aqueous methanol, and 200 µL 10 mM glucotropaeolin (benzyl glucosinolate) from PhytoPlan^®^ (Heidelberg, Germany 3403.99) was added as the internal standard (it was previously verified that this GSL was not present in any of the samples analysed). A second extraction was applied after centrifugation (5 min, 5 × 10^3^ g) using 2 mL of 70% aqueous methanol (CAS: 67-56-1). One millilitre of the combined glucosinolate extracts was pipetted onto the top of an ion-exchange column containing 1 mL Sephadex DEAE-A25 (Sigma-Aldrich, St. Louis, MO, USA, A25120) in the formiate form. Desulphation was carried out by the addition of 75 µL of purified sulphatase (E.C. 232-772-1, type H-1 from *Helix pomatia*, Sigma-Aldrich, St. Louis, MO, USA, S9751) solution. Sulphatase was purified according to the ISO protocol (ISO 9167-1, 1992). Desulphated glucosinolates were eluted with 2.5 mL (0.5 mL × 5) Milli-Q (Millipore) ultra-pure water and analysed with a Model 600 HPLC instrument (Waters) equipped with a Model 486 UV tuneable absorbance detector (Waters) at a wavelength of 229 nm. Separation was carried out using a Lichrospher 100 RP-18 in Lichrocart 125-4 column, 5 µm particle size (Merck). HPLC solvents and gradient were fixed according to the ISO protocol (ISO 9167-1, 1992). The mobile phase was a mixture of (A) acetonitrile (HPLC grade) and (B) acetonitrile/water (20:80). The flow rate was 1 mL min^−1^ in a linear gradient starting with 99% solvent A + 1% solvent B for 1 min, reaching 1% A + 99% B at 20–23 min, returning to 99% A + 1% B at 28 min, and remaining at 99% + 1% B for 10 min. The HPLC chromatogram was compared to the desulpho-glucosinolate profile of three certified reference materials, recommended by UE and ISO (CRMs 366, 190 and 367) [25], to compare the peaks with the corresponding glucosinolate. Data were corrected for UV response factors for different types of glucosinolate. The amount of each individual glucosinolate present in the sample was calculated by means of the internal standard and expressed as μmol g^−1^ of dry wt. The total glucosinolate content was computed as the sum of all the individual glucosinolate present in the sample. Data were corrected for UV response factors for different types of glucosinolates [26].

### 2.3. In Vitro Assays

The human HL-60 leukaemia cell line was grown in suspension in RPMI-1640 medium (Sigma-Aldrich, St. Louis, MO, USA, R5886), supplemented with 10% heat-inactivated foetal bovine serum (Linus, S01805), L-glutamine (200 mM, Sigma-Aldrich, St. Louis, MO, USA, G7513) and antibiotics (100IU penicillin/mL and 100 µg streptomycin/mL, Sigma-Aldrich, St. Louis, MO, USA, A5955) in a 5% CO_2_ humidified atmosphere at 37 °C. HL-60 cells were sub-cultured every 2–3 days to maintain logarithmic growth, and they were allowed to grow for 48 h before use. Cultures were plated at a density of 12.5 × 10^4^ cells/mL in 40 mL culture flasks (25 cm^2^) [27].

#### 2.3.1. Treatments

Exponential-phase HL-60 cells were placed into 12-well plates (1 × 10^5^ cells/mL, 5 × 10^5^ cells in 1.5 mL and 1 × 10^6^ cells/mL for growth inhibition, comet and DNA fragmentation assays, respectively) in triplicates and exposed to different filtered (Millipore “non-pyrogenic”, “sterile-R”, 0.2 µm filter) RPMI solutions of serially diluted samples of each *B. rapa* plant cultivar (lyophilized material prepared as described in Section 2.1) and GNA with and without myrosinase addition, and incubated under culture conditions. The GNA-myrosinase treatment was performed following a method based on that previously described for glucosinolate hydrolysis [11]. Untreated cultures were used as negative control.

#### 2.3.2. Growth Inhibition Assay

Cytotoxic activity was measured as decreased viability on treated HL-60 cells. For testing, treated and non-treated cells were counted every 24 h over 3 days in order to determine the cell growth curve. Cell viability was assessed by the Trypan Blue dye (Sigma-Aldrich, St. Louis, MO, USA, T8154) exclusion test using a hemocytometer under a light inverted microscope (AE30/31, Motic).

HL-60 graphs illustrate the sample concentration-cell response relationship with respect to the cell viability and DNA damage and distribution. Cytotoxic effect evaluation was determined during the culture treatment period, establishing a growth curve and estimating the half maximal inhibitory concentration (IC_50_) for each treatment by regression. Viability curves of leukaemia cells are presented as a survival percentage normalized to a percentage of the control at 72 h growth (control maximum exponential phase) and plotted as viability mean ± standard error of at least three independent experiments for each treatment and concentration.

#### 2.3.3. Comet Assay

The overall level of DNA damage was measured by Single Cell Gel Electrophoresis (SCGE), according to Olive and Banáth [28] and modified by ourselves [29]. For testing, 5 h treated and non-treated cells were washed twice and adjusted to 6.25 × 10^5^ cells/mL in PBS. The cell suspension was then mixed in a 1:4 dilution in low-melting temperature agarose (Sigma-Aldrich, St. Louis, MO, USA, A4018) at 40 °C and spread on a slide to form a thin electrophoresis gel. This gel was flattened out with a cover slip and gelled at RT for 30 min before taking off the cover slip. Cells were lysed in a solution of 2.5 M NaCl, 100 mM Na-EDTA, 10 mM Tris, 250 mM NaOH, 10% DMSO and 1% Triton X-100 (pH = 13) for 1 h at 4 °C and then equilibrated in alkaline electrophoresis buffer (300 mM NaOH and 1 mM Na-EDTA, pH = 13) for 20–30 min at 4 °C. After this, the slides were placed on a electrophoresis tank, exposed to an electrophoretic field of 1.25 V/cm, 400 mA for 15 min in the dark and immediately neutralized in cold neutral solution (0.4 M Tris-HCl buffer, pH 7.5) for 10 min. Slides were dried overnight at RT in the dark in order to facilitate gel evaluation. Finally, cells were stained with 7 µL propidium iodide commercial solution (Sigma-Aldrich, St. Louis, MO, USA, P4864). Fluorescence images of individual agarose-embedded cell nuclei were collected using a Leica DM2500 microscope (400× magnification) coupled with a JAI CV-M4 + CL camera and LacZ software for image acquisition.

For the analysis of comet images, at least 100 single cells from each treatment were selected at random and classified using visual scoring [30] into one of the five categories defined from 0 (no discernible tail) to 4 (ill-defined head, most DNA in tail). The category values of comets were summed, giving an overall damage score of between 0 and 400, expressed as the arbitrary units (AU) of three independent replicas in a fully balance design. One-way ANOVA followed by a Tukey’s test were used to determine significant differences between groups. Results were expressed as mean ± standard error, and *p* values below 0.05 were considered statistically significant. For the analysis of the DNA damage distribution, a Chi-square test with Yates’ correction was used, and results were expressed as number of nuclei per damage categories.

#### 2.3.4. DNA Laddering Assay

Internucleosomal fragmentation was measured to detect the proapoptotic sample’s ability to promote DNA fragmentation. For testing, 5 h treated and non-treated cells were collected and centrifuged at 3000 rpm for 5 min. For DNA extraction, a high yielding method was used [31]. Cell pellets were exposed to three solutions: (1) 900 μL of cell lysis buffer pH 8.0 (10 mM Tris-HCl, 5 mM EDTA, 100 mM NaCl), (2) 100 μL of SDS 10%, and 3) 25 μL of proteinase K solution (20 mg/mL). DNA was then incubated, shaking for 5 h at 55 °C. After this, 432 μL of 5 M NaCl was added, and samples were centrifuged at 13,000 rpm for 15 min. DNA was precipitated by adding 750 μL of cold isopropanol to each sample supernatant, centrifuged at 13,000 rpm for 10 min, washed with 1 mL of 70% ethanol, DNA dried and resuspended in 20 μL of deionised water. Finally, 0.6 μL of 0.4 mg/mL RNase was added and incubated at 37 °C (300 rpm) overnight. Total extracted DNA was quantified in a spectrophotometer (Nanodrop^®^ ND-1000), and 1200 ng of DNA were loaded into the well of an electrophoresis agarose gel (2%) and run for 2 h at 3.125 V/cm. A DNA-size ladder (GTP Bio) was run in parallel as DNA molecular weight reference. Gels were stained with ethidium bromide and digitally imaged under UV light.

### 2.4. In Vivo Assays

Two *Drosophila melanogaster* strains, each carrying a hair marker on the third chromosome, were used: (i) *mwh/mwh (mwh)*, affecting to the number of tricomas per cell on the wing surface [32], and (ii) *flr^3^/In (3LR) TM3, Bd^S^*, affecting the *flr^3^* (*flare*) marker to the tricoma shape [33]. Fly stocks and crosses were maintained at 25 °C on glass vials (2 cm diameter and 8 cm length) with a cotton cap containing a yeast-glucose medium. Transheterozygous larvae used in treatments come from the standard cross (♀ *mwh/mwh* × ♂ *flr^3^/TM3,Bd^S^*) and the reciprocal cross.

#### 2.4.1. Anti/Toxicity and Anti/Genotoxicity Assays

The Somatic Mutation and Recombination Test (SMART) [34] was used to evaluate the toxic/antitoxic and genotoxic/antigenotoxic activity of *B. rapa* cultivars (lyophylized material prepared as described in Section 2.1), as well as its selected bioactive compound GNA.

For treatments, two hundred virgin females were crossed with one hundred males, and after 8 h egg laying (72 ± 4 h later) old larvae were collected [34]. The determination of Toxicity (T) was performed following Tasset-Cuevas, et al. [35]:T = (Nº of emerging individuals in treatment/Nº of emerging individuals in the negative control) × 100(1)

Differences in *D. melanogaster* survival between simple and combined treatments at each concentration with respect to the negative and positive control, respectively, were analysed with a Chi-square test. This procedure was also performed for the survival comparison between each simple treatment and their correspondent combined treatment [31].

Genotoxicity trials were performed on groups of 100 larvae, testing serially diluted concentrations of samples: *B. rapa* cultivars and GNA.

For antigenotoxicity testing, the larvae were co-treated with the dilutions above together and the mutagen hydrogen peroxide (H_2_O_2_ 0.12 M).

Vials with the medium mixed with distilled water or H_2_O_2_ (0.12 M) were used as negative and positive controls, respectively, in both analyses.

Being treatment chronically, larvae were fed until pupation (about 48 h) and, after emergence, the resulting adult flies were sacrificed under CO_2_ narcotisation and stored in a 70% ethanol solution in sterile water. Emerged adults were counted in both simple treatments (toxicity evaluation) and combined treatment (antitoxicity evaluation). Transheterozygous wild wings (*mwh flr^3+^/mwh^+^ flr^3^*) were mounted on microscope slides and wing hair mutations (spots) scored, using a photonic microscope (Nikon) at 400× magnification for both evaluations. Transheterozygous serrate wings (*mwh/Bd^S^*) were also analysed microscopically when a treatment evaluation resulted as genotoxic.

For the evaluation of genotoxic effects, the frequencies of spots/fly of each treated series were compared to the concurrent negative control for each class of mutational clone as well as between simple and combined treatments for the same concentration comparisons. Results were categorized as positive, inconclusive or negative using a multiple-decision procedure [36]. Inconclusive and positive results were evaluated by the non-parametric *U* test of Mann, Whitney and Wilcoxon [37]. To quantify the recombinogenic rate in treatments evaluated as being genotoxic, the frequency of *mwh* clones on the marker transheterozygous individuals (*mwh* single spots plus twin spots) was compared with the frequency of mwh spots on the balancer transheterozygous wings [38].

The inhibition percentage (IP) of genotoxicity was calculated from the total frequencies of spots per wing, following Abraham [39]:IP = [(genotoxin alone − (sample + genotoxin)) × 100]/(genotoxin alone)(2)

Significant differences of IP for each treatment in respect to the positive control were analysed with a Chi-square test.

#### 2.4.2. Longevity Assay

Emerged larvae, obtained following the same procedure performed for the toxicity experiments, were used for the evaluation of the treatment effect on the longevity (lifespan) and life quality (healthspan) in the *D. melanogaster* system. In this way, this procedure assured comparable results among all the in vivo experiments.

Lifespan trials were carried out at 25 °C. Briefly, synchronized 72 ± 12 h-old trans-heterozygous larvae were washed in distilled water, collected and transferred in groups of 100 individuals into test vials containing 0.85 g *Drosophila* Instant Medium and 4 mL of the different concentrations of the compounds to be assayed. Emerged adults were collected under CO_2_ anaesthesia and placed in groups of 25 individuals of the same sex into sterile vials containing 0.21 g *Drosophila* Instant Medium and 1 mL of different concentrations of the compounds to be tested. The flies were chronically treated during their whole life. The number of survivors was recorded, and the respective nourishment renewed twice a week. In order to assure optimal *Drosophila* individual feeding with tested samples and avoid sample degradation, a specific system to renew the treated fed medium without disturbing individual development was implemented. This system consists of placing the fed medium in the vial tap and placing it inverted so it can be changed only by the substitution of the tap by another with fresh medium.

Lifespan data statistical treatment for each control and concentration was assessed by applying the Kaplan–Meier method. The significance of the curves was determined using the Log-Rank method (Mantel–Cox).

### 2.5. Statistical Analysis

All statistical analyses were performed using a Microsoft 2007 Excel spreadsheet except Tukey, Fisher, Kaplan–Meier and *U* test, which were performed with the SPSS Statistic 19.0 software (SPSS, Inc., Chicago, IL, USA).

## 3. Results and Discussion

### 3.1. Glucosinolate Profile Determination

GLS profiles and concentrations of *B. rapa* cultivars 143N5, 143N7 and 163-N7 are shown in Figure 1 and Figure 2 and Table 1 and Table 2. Significant amounts of five GLS were identified and quantified: three were aliphatic compounds (progoitrin, gluconapin and glucobrassicanapin), two were indolic compounds (glucobrassicin and neoglucobrassicin) and one was an aromatic compound (gluconasturtin).

All the cultivars have similar GLS profile but differ in their concentration, with a more than two-fold higher GLS content in cultivars 143N5 and 163N7, which presented the lowest and highest total GLS content, respectively (Table 2). Aliphatic GLS were predominant in all the cultivars, representing more than 86% of total GLS content, and especially gluconapin (GNA), which was predominant in all the *B. rapa* cultivars (from 70.73 to 88.9% of the total glucosinolate content). These results are similar to previous glucosinolate content and profile determinations for this species; Francisco, et al. [40] found a total glucosinolate content value of 26.84 µmol/g dw for turnip greens, Padilla, et al. [41] found a total glucosinolate content ranging from 11.8 to 74.0 µmol/g dw in a collection of 113 varieties of turnip greens and Cámara-Martos, et al. [42] found a total glucosinolate content value of 11.2 µmol/g dw in leaf samples of *B. rapa*.

The glucosinolate abundance profile was the same for cultivars 143N5 and 143N7, with similar percentages of each glucosinolate following the range: GNA > PRO > GST > GBN > GBS. In contrast, cultivar 163N7 presented a distinct glucosinolate proportion: GNA > GBN > GST > GBS > PRO.

In this sense, it is worth noting that progoitrin (PRO), which is considered antinutritional and described as a potentially goitrogenic glucosinolate, was the second glucosinolate in importance in samples from cultivars 143N5 (10% of total GLS) and 143N7 (12% of total GLS), and the one least abundant in samples from cultivar 163N7 (1% of total GLS). This fact could be an important key between cultivars because of the undesirable properties of PRO for human health and consumption (thyroid tumour-goitre; bitterness reducing taste preference, respectively) [17,43]. However, there is no evidence for any goitrogenic effect on humans from *Brassica* consumption [16,18], so more studies related to individual glucosinolates ought to be undertaken in order to elucidate their effect.

### 3.2. Activity Against Tumour Cell Proliferation

#### 3.2.1. Cytotoxicity

Cell survival determination was performed as a preliminary range-screening experiment to determine sample cytotoxicity. In this sense, viable cells were not found when applying *B. rapa* samples at concentrations of over 2 mg/mL (Figure 3a–c). Regarding *B. rapa* cultivars, total glucosinolate content was correlated with cytotoxicity so that cultivar 143N5 presented a higher IC_50_ value than 163N7. The effect of cell viability reduction was stronger in the case of hydrolysed GNA samples, which completely prevented cell growth at a concentration of around 0.01 mg/mL (Figure 3d). However, cell parallel exposure to non-hydrolysed GNA was found to be ineffective in the ability of this glucosinolate to inhibit cell growth (Figure 3e).

We have previously reported this glucosinolate hydrolysis-dependent effect in close related *Brassica* spp. [11]. This digestion process could be accomplished only once the myrosinase enzyme was added to the GNA treatment solution. Our results agree with those above and confirm that the break-down products of glucosinolate hydrolysis are the ones responsible for this anti-tumour activity. It is worth noting that the inhibition curves of entire plants mimic the one for GNA + myrosinase, which would mean that GNA content is the main factor in charge of the *B. rapa* entries cytotoxic activities.

#### 3.2.2. DNA Damage

The comet assay was selected in this work in order to monitor the DNA changes that brought about dietary intervention with glucosinolates by the measurement of genomic DNA strand-breaks on individual cells. This method is not only simple, economic, reliable, versatile, rapid and widely-used for DNA damage measurement, but it is also ideally suited for testing genotoxicity in suspension cell cultures [44]. However, it is important to take into account the technique requirements in order to obtain valuable results. Because the comet assay is based on the visualization of individual nuclei, it needs the presence of cells into samples. Thus, for the treatments, we selected a range of sample concentrations that allowed cell growth in cytotoxicity experiments. Additionally, due to the presence in some treatments of a considerable amount of heavily damaged comets that computer programs are unable to monitor [45], we decided to use a scoring by naked eye as a more appropriate method for DNA damage measurement. Representative images of the range of different comet categories are shown in Figure 4. Both scoring methods have been proved to produce close results that, in practice, are virtually indistinguishable [46]. After these considerations, our experiments showed that the DNA break induction in all treatments was significantly different from that of untreated cells (Figure 5).

With regard to the samples, a dose-effect was observed in *B. rapa* 143N7 and 163N7 as well as in the hydrolyzed GNA treatments. Moreover, the highest assayed doses of these treatments resulted in the highest levels of DNA damage. Contrarily, *B. rapa* 143N5 and GNA treatments showed the lowest capability to induce DNA strand-breaks on cells independently of the assayed dose. Besides, data statistical analysis grouped these two treatments as being the least harmful ones. Other authors have used this method for testing the healthy properties of widely consumed close-related members of *B. rapa*, i.e., broccoli, Brussels sprouts, kale [47,48,49,50], always supporting the consumption of *Brassica* vegetables for being chemopreventive agents.

Analysed comet images also provided information on how treatments affected individual cells, revealing the sample DNA damage distribution pattern (Figure 6).

In this sense, treated cells from all samples showed a DNA damage distribution pattern significantly different from untreated cells. Control cells were characterized by the presence of nuclei classified mainly into the two lowest categories of DNA damage (0 and 1), as expected. On the contrary, nuclei induced by *B. rapa* 143N7 and 163N7, and hydrolyzed GNA samples were mainly classified into more damaged categories (3 and 4). This fact was more evident in the highest concentration of these treatments in which almost all the nuclei analysed were included in the greater damage category (4). Additionally, we were able to observe a DNA damage–sample dose relationship, so DNA damage reduced together with sample concentration. In addition, comets moved homogeneously through the damage categories in *B. rapa* treatments as the plant sample concentration increased. In the case of hydrolyzed GNA, this effect was more abrupt, with almost all comets being exposed to intermediate, and the highest concentrations evaluated as being the most damaged ones (category 4) and classified into different damage categories from the majority of comets in the lowest concentration treatment. On the other hand, nuclei in *B. rapa* 143N5 and GNA treatments did not show that DNA damage pattern.

#### 3.2.3. DNA Fragmentation

DNA damage can be the result of different cell death mechanisms, such as necrosis or apoptosis, and to distinguish between them is of a biological relevance in cancer therapy [51]. Because apoptosis implies genomic DNA cleavage to the size of oligonucleotides, comet analysis is unable to detect this process [44,52]. For this reason, we decided to examine the capacity of tested samples to promote DNA fragmentation qualitatively by conventional constant field agarose gel electrophoresis. With this method, the apoptosis process was recognized by the appearance of internucleosomal DNA fragments that are multiples of 200 base pairs (Figure 7).

Gels observation revealed extensive DNA fragmentation to nucleosome size, corresponding to cells treated with all the *B. rapa* cultivars at all the concentrations assayed (Figure 7a–c). In addition, it was observed that these treatments produced concentration-dependent increases in DNA fragment production. On the other hand, exposure to intact GNA failed to induce DNA fragmentation of the sample concentration independently (Figure 7e). Contrarily, after myrosinase hydrolysis, DNA fragments were visible at all evaluated concentrations in the same way as the plant treatments but showed a threshold at the concentrations assayed (Figure 7d). This fact again means that the breakdown products of GNA hydrolysis are responsible for this action, not the GNA itself. These results were consistent with our previous in vitro experiments performed above.

### 3.3. Activity Against Degenerative/Oxidative Processes

*Drosophila* organism is a valuable eukaryotic system for toxicity evaluation due to its high percentage of homologue genes in common with humans [53]. In this regard, the *Drosophila* system has been recently presented as an advantageous model to study the genetic causes of variation in toxicity susceptibility in order to guide studies in human populations [54] and to support its extensive use in toxicity tests [55,56].

Thanks to the results obtained for the different treatments previously performed in vitro assays and in order to optimize analyses, for the in vivo experiments we selected *B. rapa* cultivars 143N5 and 163N7 as the most interesting samples. Additionally, because previous studies showed that the digestion process of *Drosophila* individuals results in the appearance of GSL bioactives [11], a GNA + myrosinase treatment is not required to analyse the GSL breakdown product effect. For that reason, *Drosophila* individuals were directly exposed to the intact GSL.

#### 3.3.1. Anti/Toxicity Studies

The effects of *B. rapa* and GNA treatments in *Drosophila* survival is showed in Figure 8, expressed as the percentage of emerged (survival %) treated adults with respect to the negative control emerged adults (survival control corrected).

Differences between *B. rapa* cultivars were observed in the simple treatments (Figure 8a), with 163N7 treatments being less toxic than those of 143N5. In this sense, the highest *B. rapa* 163N7 concentration did not affect *Drosophila* survival, while the intermediate *B. rapa* 143N5 concentration was the most toxic treatment, reducing larvae survival by around 50%. However, no dose effect appeared in the *B. rapa* treatments. Conversely, GNA treatments resulted in a marked dose effect from 61 to 96% survival (lowest and highest assayed concentrations, respectively) but only the highest concentration was over the toxicity limit. It is of interest that the 143N5 intermediate and highest concentrations reduced *D. melanogaster* survival, exceeding the toxicity limit. This could be explained by the GSL profile of this cultivar, which presented a different pattern with respect to that of cultivar163N7.

Sample ability to counteract the damage induced by H_2_O_2_ has been demonstrated in our experiments, in which almost all the treatments offset the individual survival reduction by this toxicant throughout the concentration ranges assayed (Figure 8b). When comparing simple and combined treatments, we observed that the H_2_O_2_ addition to the feed medium significantly reduced *D. melanogaster* survival, except in some *B. rapa* treatments, highlighting the protective effect of this plant in feeding and its role as a reactive oxygen species scavenger. Otherwise, the survival effect in GNA treatments was significant at all the concentration assayed, but the highest concentration was the only one that reached the anti-toxicity limit. This fact confirms the dose-dependent action of GNA as an anti-toxicity agent, requiring a specific concentration to exert this health benefit to health functions. Considering these results, it can be inferred that, during the *Drosophila* feeding process of the GNA treatment, hydrolysis occurs as expected, leading to the appearance of the GNA derivative effect.

#### 3.3.2. Longevity and Healthspan Studies

Fleming et al. [57] have investigated the oxidative stress during *Drosophila* ageing. They found that the production of reactive oxygen species is directly correlated with *Drosophila’s* physiological decline. In this sense, lifespan bioassays have recently turned out to be an excellent method for elucidating the relationship between life expectancy and dietary habits.

All three treatments dramatically affected survival at all concentrations, exhibiting increases in lifespan of 20.2, 31.4 and 14.2%, on average, when *Drosophila* was fed chronically with *B. rapa* cultivars 143N5 and 163N7 and GNA, respectively (Table 3). The highest content in GNA of cultivar 163N7 could be related to the highest increases in lifespan in most concentrations. For GNA treatments, a negative dose response effect is observed, with lower increases at the highest concentrations.

Positive increases in healthspan are also observed in all the concentrations (Figure 9), although significant values are mainly observed in 163N7 with double and fourfold the survival rates. GNA does not influence healthspan extension in *Drosophila melanogaster*.

Those *Brassica rapa* cultivars with a high content in glucosinolate, to be precise, GNA molecule, showed a strong life extension activity in the present research. Nevertheless, no references in this respect have been found. The only supporting data come from similar experiments carried out using sulforaphane isothiocyanate (the hydrolysis by-product from the breakdown of glucoraphanin glucosinolate) and *Eruca sativa* whole plant [58]. Comparisons can be made with them, as hydrolysed GNA in *Drosophila* feeding and metabolism is understood. Sulforaphane was not able to increase lifespan, nor were the *Eruca* extracts, which showed only increases in lifespan curves. Hence, with respect to the effect on lifespan, the *B. rapa* entries selected can be considered as a promising food to be used as a nutraceutical element, i.e., the glucosinolate content in entry 163N7.

#### 3.3.3. Anti/Genotoxicity Assays

Currently accepted hypotheses propose antioxidant properties as being the origin of the *B. rapa* health benefits [10]. It could be assumed that the best way to confirm this antioxidant property would be to compare *B. rapa* plant consumption to an oxidant molecule. For this reason, we have selected H_2_O_2_ as an oxygen free radical generator that affects *D. melanogaster* development, inducing mutations that are easily scored by microscopy observation [59]. The use of H_2_O_2_ as a positive control would be an acknowledged method to prove antioxidant activity.

Results obtained from genonotoxicity and antigenotoxicity experiments are summarized in Table 4 and Table 5, respectively, as total mutations (spots) per wing in *Drosophila* adults exposed to *B. rapa* and GNA treatments.

Our results showed that the only genotoxic treatment was the lowest concentration of *B. rapa* 143N5 assayed that produced 0.425 mutations per wing. Conversely, the lowest concentration of *B. rapa* 163N7 and GNA presented the lowest rate of mutations per wing, with the same value as the negative control (0.175 spots/wing). Being a healthy food, edible vegetables are rarely related to genotoxic effects [60,61]. Surprisingly, the assayed *B. rapa* 143N5 cultivar was evaluated as being genotoxic. However, this genotoxicity was not related to a vegetable species but with a selected crop cultivar with a different secondary metabolite content.

No studies on *B. rapa* genotoxicity have been performed to date due to the well-known health promoting effects of this plant and cruciferous vegetable [62]. The results in our work not only confirm this fact but also prove the important role of *B. rapa* phytochemicals.

To estimate the recombinogenic potency of mutagenic samples, we searched for the additional information on the spots per wing scored in balancer wings (Serrate phenotype) (Table 4). In the balancer-heterozygous genotype (*mwh/TM3, Bd^S^*), mwh spots are mainly produced by somatic point mutation and chromosome aberrations because mitotic recombination between the balancer chromosome and its structurally normal homologue is a lethal event. The difference in mwh clone frequency is a direct measurement of the proportion of recombination [38]. The recombinogenicity value with respect to the total induced clones was 37.5% for *B. rapa* 143N5 treatment, which was close to the value obtained for the H_2_O_2_ treatment (42.1%). This would indicate that the *B. rapa* 143N5 treatment exerts its genotoxic effect mainly by somatic point mutation rather than by mitotic recombination.

An antigenotoxicity test revealed the treatment capability to counteract the H_2_O_2_ genotoxic effect and its potential to protect DNA from H_2_O_2_ damage. Our results (Table 5) have demonstrated that the treatments were able to eliminate part of this genotoxic effect, even reducing the apparition of mutations to a greater degree than negative control. This was the case for the highest concentration of *B. rapa* 163N7 and GNA treatments (0.147 and 0.128 spots/wing, respectively) that resulted in inhibition percentages of genotoxicity of 69 and 73%.

In this sense, cruciferous vegetable consumption is strongly recommended due to their important antioxidant activity [63,64], being the main anticancer food because of their abundant antioxidants as GNA. Our results are according to this affirmation and maintain the idea to include selected *B. rapa* cultivar consumption in our diet.

## 4. Conclusions

Our results stress the importance of the content in plant secondary metabolites when aiming at a healthy diet.

A positive relationship between the glucosinolate content and the biological activities has been found for the cultivars selected. A first screening of in vitro HL-60 cytotoxicity assays gave similar positive results for the three cultivars (143N5, 143N7 and 163N7). All of them showed high level of internucleosomal DNA fragmentation and hedgehog-like comet DNA single/double strand clastogenic activities, as well as a similar IC_50_. As a consequence, for the more time-consuming in vivo bioassays, the 143N5 and 163N7 entries were selected as they showed the most different GSL profile, both in total content and in PRO and in GBN relative contents (see Table 2). From an overall evaluation, taking into account the quantitative and qualitative results of the in vitro and in vivo trials, several conclusions can be drawn.

From the food safety point of view, the cultivar 143N5 would not be a recommended choice as a human food for frequent consumption as it was mutagenic in the somatic mutation and recombination test. In addition, assigning a categorical value to result in the different assays in order to compare the two selected cultivars, the 163N7 results as the cultivar of choice as it occupies the first position in most assays: cytotoxicity, fragmentation, comet, toxicity, antitoxicity, longevity, healthspan, genotoxicity and antigenotoxicity. Alternatively, the 143N5 is the first *ex aequo* with 163N7 in comet and antitoxicity, but it is mutagenic in the SMART test.

Cultivar 163N7, selected at the Institute for Sustainable Agriculture (Córdoba, Spain) for having the highest total glucosinolate content and lowest progoitrin content, showed itself to be safe (non-toxic and non-genotoxic). More importantly, it was seen as a real nutraceutical proposal due to its genomic protective activity (antigenotoxic), lifespan extension, and chemopreventive activities against leukaemia cells. We suggest including it among the *Brassica* health promoters.

## Figures and Tables

**Figure 1 foods-10-02720-f001:**
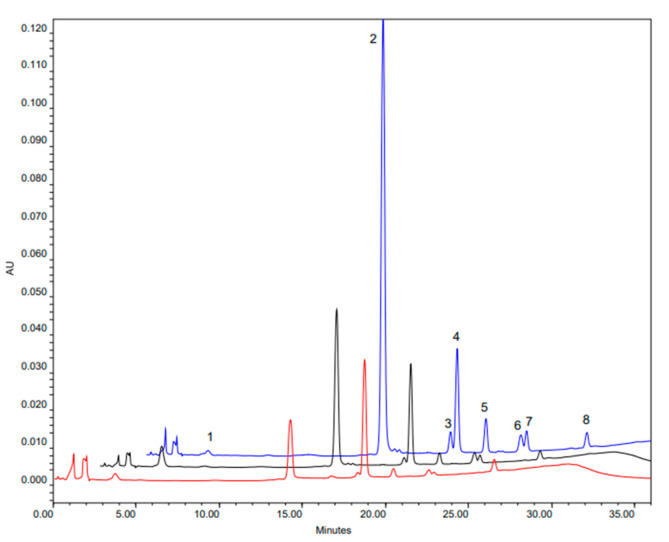
HPLC chromatogram of glucosinolate profile of turnip greens. Varieties: 143N5 (red), 143N7 (black) and 163N7 (blue). Peaks: (1) Progoitrin, (2) Gluconapin, (3) Glucobrassicanapin, (4) Glucotropaeolin (internal standard), (5) Glucobrassicin, (6) 4-Metoxyglucobrassicin, (7) Gluconasturtin, (8) Neoglucobrassicin.

**Figure 2 foods-10-02720-f002:**
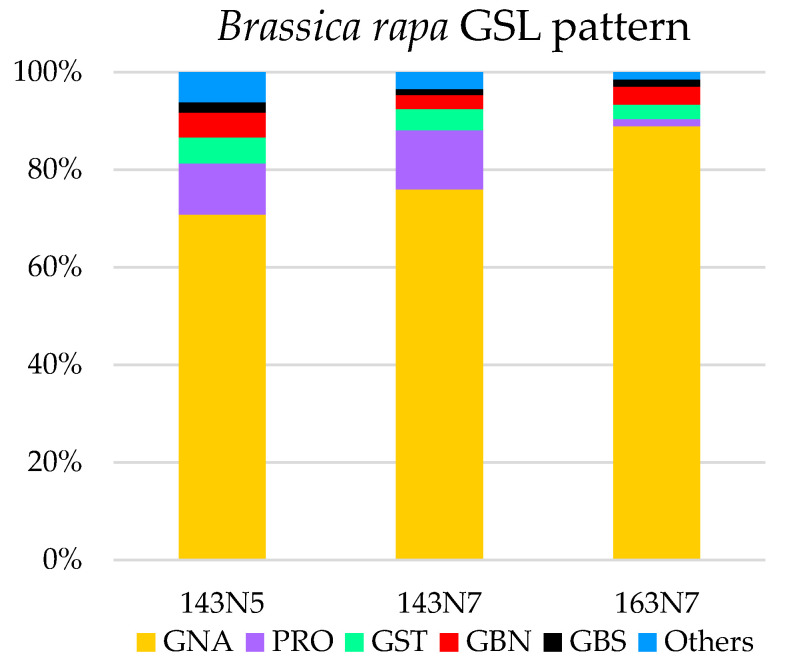
Glucosinolate percentage in analysed *B. rapa* cultivars.

**Figure 3 foods-10-02720-f003:**
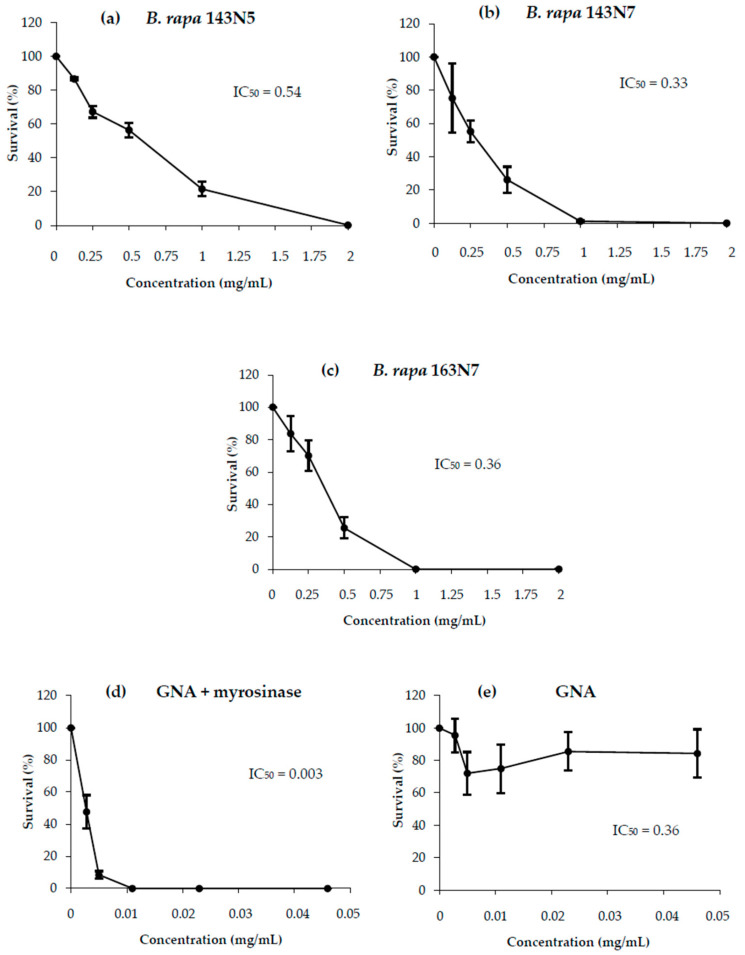
Survival of HL-60 cultures treated with different concentrations of: *B. rapa* (**a**) 143N5, (**b**) 143N7 and (**c**) 163N7 plant material; bioactive compound: (**d**) hydrolyzed gluconapin and (**e**) gluconapin. Survival curves are plotted as percentages with respect to the control counted at 72 h treatment from at least three independent experiments (mean ± SE).

**Figure 4 foods-10-02720-f004:**
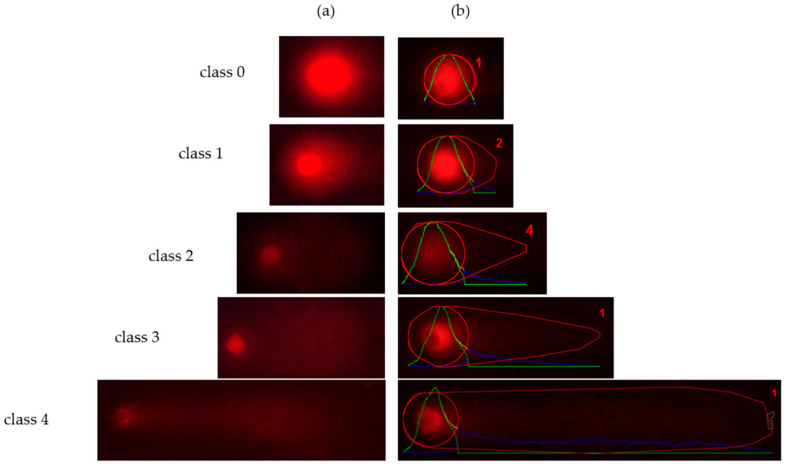
Representative images of the range of different comet categories. (**a**) Photographs obtained in our comet assay for each treatment. (**b**) Tail Moment (TM) parameter analysed using OpenComet plugging for ImageJ Software. Class 0: TM < 1 (negative control—no damage); class 1: TM = 1–5 (shot tails—slightly damaged); class 2: TM = 5–10 (medium damaged); class 3: TM = 10–20 (hedgehog pattern—highly damaged); class 4: TM > 20 (completely damaged—apoptosis).

**Figure 5 foods-10-02720-f005:**
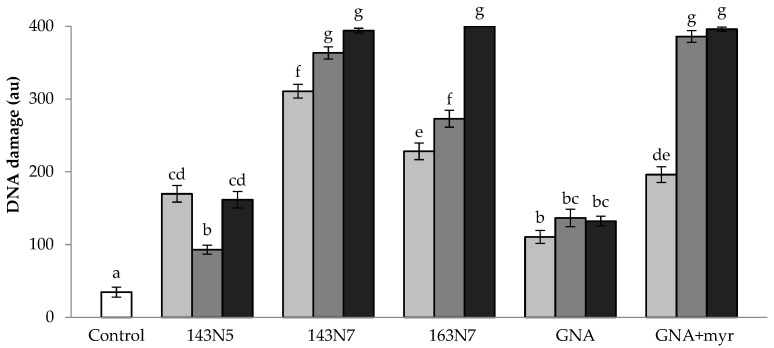
The level of DNA damage in treated HL-60 cells. Greyscale represents increasing concentrations of *Brassica rapa* (0.31, 0.625 and 1.25 mg/mL) and hydrolyzed and non-hydrolyzed gluconapin (0.0069, 0.0137 and 0.025 mg/mL) samples, the white column being the non-treated ones (control). Columns with the same letters do not differ at a level of 5% significance on the Tukey’s test.

**Figure 6 foods-10-02720-f006:**
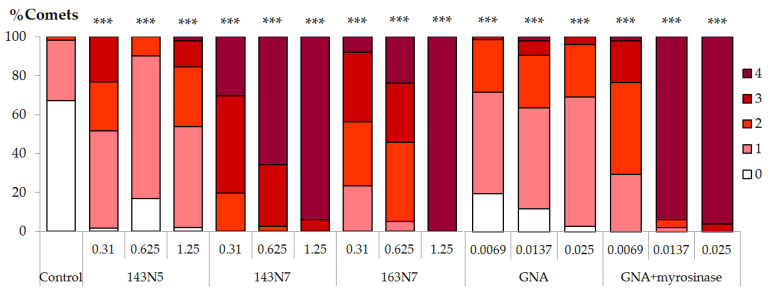
Comet distribution into visual scoring categories in HL-60 cells treated with increasing concentrations of *B. rapa*, gluconapin and hydrolyzed gluconapin samples (mg/mL). Categories are shown using a red colour scale from white (category 0) to deep red (category 4). *** Significance levels with respect to control (*p* ≤ 0.001).

**Figure 7 foods-10-02720-f007:**
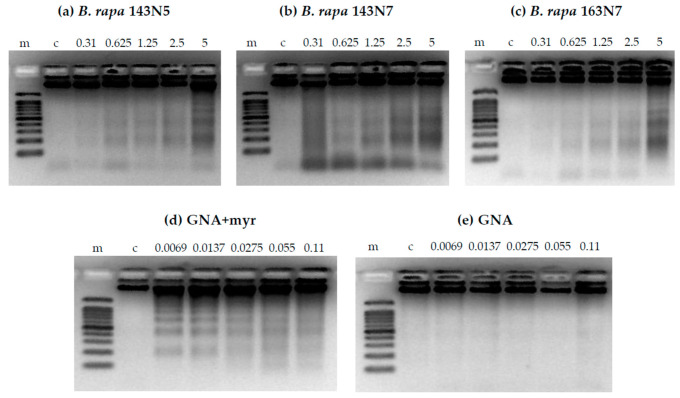
Genomic DNA degradation in HL-60 cells exposed to increasing concentrations of: (**a**–**c**) *B. rapa* (0.31, 0.625, 1.25, 2.5 and 5 mg/mL), (**d**) hydrolyzed and (**e**) non-hydrolyzed gluconapin (0.0069, 0.0137, 0.0275, 0.055 and 0.11 mg/mL) samples. 100 pb DNA ladder, PROMEGA (m) was electrophoresed as a base pair reference (Lane 1) and untreated cells (**c**) as control (Lane 2).

**Figure 8 foods-10-02720-f008:**
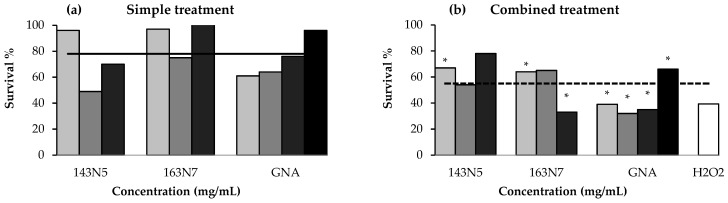
*Drosophila* survival in simple (**a**) and combined (**b**) treatments expressed in percentage as total emerged adults of each treatment with respect to the negative control (H_2_O) total emerged adults. Combined treatments were performed adding standard medium and 0.12 M H_2_O_2_ (positive control). Grey scale represents increasing concentrations of *B. rapa* (1.25, 2.5 and 5 mg/mL) and gluconapin (0.0137, 0.0275, 0.055 and 0.11 mg/mL) samples, the white column being the positive control (H_2_O_2_). Toxicity and antitoxicity limits (black and dashed lines, respectively) represent the significance toxicity and antitoxicity levels with respect to the negative and positive controls, respectively (*p* ≤ 0.025). * Significance levels between simple and combined treatment for the same concentration comparisons (*p* ≤ 0.05).

**Figure 9 foods-10-02720-f009:**
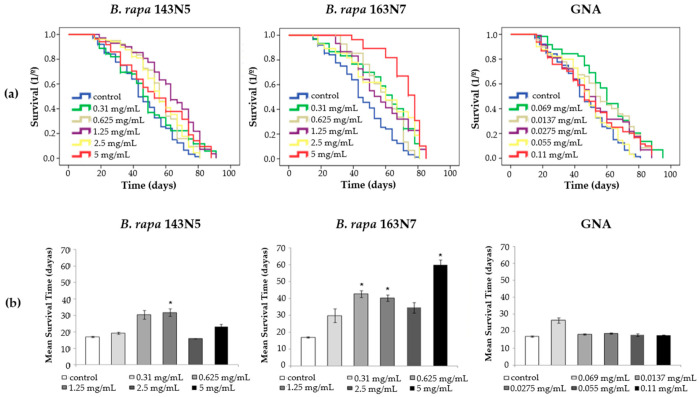
Effect of *B. rapa* cultivars 143N5 and 163N7 and gluconapin (GNA) supplementation on *D. melanogaster* survival: (**a**) Survival curves; (**b**) Healthspan averages (the mean of survival time, in 75% of surviving population, is shown for each concentration and sample). * significant (*p* < 0.05) with respect to their control.

**Table 1 foods-10-02720-t001:** Trivial name, chemical class, systematic name and abbreviations of major glucosinolates found in leave samples of *Brassica rapa* cultivars.

Trivial Name	Chemical Class—Systematic Name	Abbreviation
	Aliphatic	
Gluconapin	3-Butenyl glucosinolate	GNA
Progoitrin	2-Hydroxy-3-butenyl glucosinolate	PRO
Glucobrassicanapin	4-Pentenyl glucosinolate	GBN
	Aromatic	
Gluconasturtin	2-Phenylethyl glucosinolate	GST
	Indole	
Glucobrassicin	3-Indolymethyl glucosinolate	GBS

**Table 2 foods-10-02720-t002:** Glucosinolate content in analysed *Brassica rapa* cultivars.

Glucosinolate Content (µmol/g Dry Weight)							
			Aliphatic		Indole	Aromatic	
*B. rapa*	Total	GNA *	PRO	GBN	GST	GBS	Others
143N5	8.58	6.07 (1)	0.90 (2)	0.43 (4)	0.46 (3)	0.19 (5)	0.53
143N7	25.45	19.35 (1)	3.06 (2)	0.72 (4)	1.12 (3)	0.31 (5)	0.89
163N7	60.88	54.12 (1)	0.89 (5)	2.21 (2)	1.82 (3)	0.92 (4)	0.91

* Abbreviation of GLS trivial name. See Table 1. Numbers in brackets represent the abundance of each individual GSL in each plant cultivar from 1 (most abundant) to 5 (less abundant).

**Table 3 foods-10-02720-t003:** Effects of *B. rapa* 143N5 and 163N7 cultivars and gluconapin (GNA) treatments on the *D. melanogaster* average lifespan and healthspan (75%).

Compound	Concentration (mg/mL)	Lifespan Average (days)	Mean Lifespan Difference (%)	Healthspan Average (days)	Mean Healthspan Difference (%)
*B. rapa* 143N5	control	46.365		16.929	
	0.31	49.506	7	19.231	13
	0.625	56.332 **	21	30.308	79
	1.25	62.747 ***	35	31.714 *	87
	2.5	55.891 **	20	16.000	−5
	5	54.611 *	18	23.286	37
*B. rapa* 163N7	control	46.365		16.929	
	0.31	58.098 **	25	40.238	137
	0.625	58.510 *	26	42.679 *	152
	1.25	57.908 **	26	40.302 *	138
	2.5	57.917 ***	25	34.574	104
	5	72.066 ***	55	59.854 *	253
GNA	control	46.365		16.929	
	0.0069	59.931 ***	29	26.357	55
	0.0137	55.157 ***	19	18.143	7
	0.0275	51.819 **	12	18.700	10
	0.055	48.241	4	17.929	6
	0.11	49.828 *	7	17.500	3

Means were calculated by the Kaplan–Meier method, and significance of the curves was determined by the Log-Rank method (Mantel–Cox). *: significant (*p* < 0.05), **: highly significant (*p* < 0.01), ***: very highly significant (*p* < 0.001).

**Table 4 foods-10-02720-t004:** Genotoxicity of *Brassica rapa* (Br) cultivars and gluconapin (GNA) in the *Drosophila* wing spot test.

Mutation Rate (Spots/Wing) Diagnosis ^1^						
Compound	N° of Wings	Small Spots (1–2 Cells) m = 2	Large Spots (>2 Cells) m = 5	Twin Spots m = 5	Total Spots m = 2	Estimate Recombination Percentage
H_2_O	40	0.15 (6) i	0.025 (1) d	0	0.175 (7)	
H_2_O_2_ (0.12 M)	40	0.3 (12) i	0.15 (6) i	0.025 (1) d	0.475 (19) +	
H_2_O_2_ (0.12 M) (S ^2^)	40	0.275 (11) +	0	0	0.275 (11) +	[(0.475–0.275)/0.475] × 100 = 42.107
*B. rapa* 143N5 (mg/mL)						
1.25	40	0.375 (15) i	0.05 (2) −	0	0.425 (17) +	
1.25 (S ^2^)	40	0.225 (9) +	0.025 (1) i	0	0.25 (10) +	[(0.4–0.25)/0.4] × 100 = 37.5
5	40	0.2 (8) i	0.025 (1) d	0.05 (2) −	0.275 (11) i	
*B. rapa* 163N7 (mg/mL)						
1.25	40	0.175 (7) i	0	0	0.175 (7) i	
5	40	0.125 (5) i	0.075 (3) −	0.025 (1) d	0.225 (9) i	
GNA (mg/mL)						
0.0137	40	0.1 (4) −	0.05 (2) −	0.025 (1) d	0.175 (7) i	
0.11	40	0.25 (10) i	0	0	0.25 (10) i	

^1^ Statistical diagnoses as stated by Frei and Würgler [36,37]: + (positive) and − (negative). Significance levels α = β = 0.05, one-sided test without Bonferroni correction. ^2^ Balancers-heterozygous (Serrate) wings. Number of spots is showed in brackets.

**Table 5 foods-10-02720-t005:** Antigenotoxicity of *B. rapa* cultivars and gluconapin (GNA) in the *Drosophila* wing spot test.

Mutation Rate (Spots/Wing) Diagnosis ^1^						
Compound	N° of Wings	Small Spots (1–2 Cells) m = 2	Large Spots (>2 Cells) m = 5	Twin Spots m = 5	Total Spots m = 2	IP *^2^* (%)
H_2_O	40	0.15 (6)	0.025 (1)	0	0.175 (7)	
H_2_O_2_ (0.12 M)	40	0.3 (12) i	0.15 (6) i	0.025 (1) d	0.475 (19) +	
*B. rapa* 143N5 (mg/mL)						
1.25	40	0.225 (9) i	0	0.025 (1) d	0.25 (10) i	47.4 *
5	40	0.15 (6) i	0.025 (1) d	0	0.175 (7) i	63.2 **
*B. rapa* 163N7 (mg/mL)						
1.25	40	0.175 (7) i	0.025 (1) d	0.025 (1) d	0.225 (9) i	52.6 *
5	34	0.147 (5) i	0	0	0.147 (5) i	69 **
GNA (mg/mL)						
0.0137	40	0.2 (8) i	0	0.025 (1) d	0.225 (9) i	52.6 *
0.11	39	0.128 (5) i	0	0	0.128 (5) i	73 ***

^1^ Statistical diagnoses as stated by Frei and Würgler [36,37]: + (positive) and − (negative). Significance levels α = β = 0.05, one-sided test without Bonferroni correction. ^2^ Strength of inhibition on the capability of H_2_O_2_ (0.12 M) to induce mutated cells (Inhibition Percentage). * Significance levels with respect to the positive control (H_2_O_2_) group (* *p* ≤ 0.05; ** *p* ≤ 0.01; *** *p* ≤ 0.001). Number of spots is shown in brackets.

## Data Availability

The datasets generated during the current study are available from the corresponding author on reasonable request.

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
