# Peer review of "Role of Glucosinolates in the Nutraceutical Potential of Selected Cultivars of Brassica rapa"

_foods, 2021, doi:10.3390/foods10112720_

Round 1

Reviewer 1 Report

Dear Author

The requested modifications have been reported in the attached pdf file (text).

Author Response

Please, find attached the reply's document.

Reviewer 2 Report

The authors of the manuscript ‘Role of glucosinolates in the nutraceutical potential of selected cultivars of Brassica rapa’ reported the relation between B. rapa glucosinolates content and their beneficial properties. The identification and quantification of five glucosinolates by HPLC analysis in leaves of three B. rapa cultivars have also been evaluated. The results reported are encouraging and will be helpful in the selection of new B. rapa cultivar with health benefits. Although the study is essential, the manuscript suffers from several drawbacks, which are given below:

L12: Brassica rapa L. subsp. rapa instead of Brassica rapa L.

L27: turnip greens instead of Brassica rapa

L34: Brassicaceae – nonitalic.

L42: Brassicaceae – nonitalic.

L61: Brassicaceae – nonitalic.

L61: (GLS) instead of (glucosinolates).

L62: GLS instead of glucosinolates.

L70: subsp. – nonitalic.

L83: Add a note on studied cultivars.

L85: The objective of this investigation is not written.

L105: What about the other four standards of GLS.

L112: HPLC columns, solvents and gradient were fixed according to the ISO protocol. Please provide complete procedures.

L116: Te (typo).

L117: Please indicate the studied glucosinolates.

L312: ‘All the cultivars have similar GLS profile but differ in its concentration’: For better understanding, please provide HPLC chromatograms of leaf extracts of studied cultivars.

L317-318: These results are similar to previous glucosinolate content and profile determinations for this species [39-41]. Please indicate the levels of glucosinolate for better understanding.

L385: Correct the Figure 2 alignment.

L398: 3.2.2. DNA damage - Comet assay representative images are required.

Author Response

(The authors gave the same response as above.)

Round 2

Reviewer 1 Report

Dear Authors

Thanks for your reply to my commnets

I think now your manuscript is perfectly modified

it remains only few minor points to check.

COMMENT 3.- Line 18

(...........properties in vivo and in vitro toxicity)

R- please start with "in vitro" then "in vivo".

-------------------------------------------------

COMMENT 5.- Lines 36 y 37.

R- the name of family should be written in Italic "Brassicaceae".

-----------------------------------------------------

COMMENT 8.- Line 50. what does it refer to ( GLS or products)?

R- it is clear that the chemopreventive activity of glucosinolates is related to the metabolic breakdown products of GLS.
hence in my comment i asked just to link  this phrase to the previous original one.
I think that you can say ( Some GLS are healthy chemopreventive active while others are harmful since they are potentially goitrogenic).

---------------------------------------------------

COMMENT 13.- Lines 99 to 100. 

Thanks for the authors for the explanation
i think that you can mention the details of the extraction in a separate paragraph then in the both assays (in vitro assays and in vivo) you can only refer to the prepared or extacted materials as previouly mentioned. 

----------------------------------------

COMMENT 15.- Lines 236 to 238. Statistical analysis
I agree with the authors that statistical section can be spliced into the different M&M sections. if you used different statistical program , then ok, you can leave it as it was written in the original version.

----------------------------------------

COMMENT 18.- Line 344. (inhibited or degraded).
in this case i would like to point to the term "non-viable" means that the cells remain not degraded but became not able to reproduced as normaly. so we can say inhibited their reproduction . But if you mean properly degraded so there are no need to say "non-viable".  please check this part again.

----------------------------------------

Lines:  

.....................The consumption of plant-based foods with nutraceutical properties is one of the crucial factors contributing to well-being, and to the promotion of health) 

My commnet:  please cite the following reference in this part of introduction related the pharmaceutical properties of vegetal extracts, i think this is important to address general information before specify on (Brassicaceae family):

Elshafie, H.S.; et al   2021.   Plants 2021, 10, 153. https://doi.org/10.3390/plants100101

Best regards

Author Response

ANSWERS TO REVIEWER 1. R2

COMMENT 3.- Line 18

(...........properties in vivo and in vitro toxicity)

R- please start with "in vitro" then "in vivo".

ANSWER:

Thanks, the sentence has been modified.

-------------------------------------------------

COMMENT 5.- Lines 36 y 37.

R- the name of family should be written in Italic "Brassicaceae".

ANSWER:

There is a general consensus that the name of the genus and species of a plant should be written in italics, but there are different opinions on how to write the name of the plant family, Brassicaceae in our case. In the first version of the manuscript we used the name in italics. Following the recommendation of reviewer 2 we eliminated the italicization. In journals such as the American Journal of Botany, italics are not used to name the family Brassicaceae (see reference https://bsapubs.onlinelibrary.wiley.com/doi/full/10.3732/ajb.93.4.607). In the International Code of Nomenclature for algae, fungi, and plants (Shenzhen Code) adopted by the Nineteenth International Botanical Congress Shenzhen, China, July 2017 it is recommended to use italics for naming plant families.

Following the reviewer's recommendation we have reitalicized the family name Brassicaceae

-----------------------------------------------------

COMMENT 8.- Line 50. what does it refer to ( GLS or products)?

R- it is clear that the chemopreventive activity of glucosinolates is related to the metabolic

breakdown products of GLS.

hence in my comment i asked just to link this phrase to the previous original one.

I think that you can say ( Some GLS are healthy chemopreventive active while others are harmful since they are potentially goitrogenic).

ANSWER:

Thanks. The sentence has been modified following the advice of the reviewer.

---------------------------------------------------

COMMENT 13.- Lines 99 to 100.

Thanks for the authors for the explanation

i think that you can mention the details of the extraction in a separate paragraph then in the both assays (in vitro assays and in vivo) you can only refer to the prepared or extacted materials as previouly mentioned.

ANSWER:

In accordance with the reviewer's recommendation, we have included a new paragraph in sections 2.3.1 and 2.4.1 specifying that the samples of lyophilized material used were obtained according to the procedure described in section 2.1.

----------------------------------------

COMMENT 15.- Lines 236 to 238. Statistical analysis

I agree with the authors that statistical section can be spliced into the different M&M sections. If you used different statistical program , then ok, you can leave it as it was written in the original version.

ANSWER:

Authors consider that both ways for describing statistical methods are equally understandable. Nevertheless, we consider that the first suggestion of the reviewer is an easy reading way to show the different statistical methodologies used.

----------------------------------------

COMMENT 18.- Line 344. (inhibited or degraded).

in this case i would like to point to the term "non-viable" means that the cells remain not degraded but became not able to reproduced as normaly. so we can say inhibited their reproduction . But if you mean properly degraded so there are no need to say "non-viable". please check this part again.

ANSWER:

 The trypan blue assay is able to detect both viable (non-coloured cells) and the so called non-viable cells (blue coloured cells which membrane are not able to exclude the colorant). Therefore, under the name nonviable cells we include all different cellular situations corresponding to coloured cells. Being our scientific interest focused on the effect of the tested substances on viable tumoural cells, we consider that the terminology viable/non-viable fit to describe the biological cell state on treatment.

----------------------------------------

Lines:

.....................The consumption of plant-based foods with nutraceutical properties is one of the

crucial factors contributing to well-being, and to the promotion of health)

My commnet: please cite the following reference in this part of introduction related the

pharmaceutical properties of vegetal extracts, i think this is important to address general

information before specify on (Brassicaceae family):

Elshafie, H.S.; et al 2021. Plants 2021, 10, 153. https://doi.org/10.3390/plants100101

Best regards

ANSWER:

Thanks for this suggestions. The reference has been included in the introduction section.

Reviewer 2 Report

The manuscript has been significantly improved.

Author Response

Authors thanks the reviewer's comments.

Spell has been checked in the manuscript.